# Association between Self-Perceived Periodic Limb Movement during Sleep and Excessive Daytime Sleepiness Depend on Restless Leg Symptoms in Korean Adolescents

**DOI:** 10.3390/ijerph19084751

**Published:** 2022-04-14

**Authors:** Hyeyun Kim, Kwang Ik Yang, Jun-Sang Sunwoo, Jongkyu Park, Nam Hun Heo, Jee Hyun Kim, Seung Bong Hong

**Affiliations:** 1Department of Neurology, International St. Mary’s Hospital, College of Medicine, Catholic Kwandong University, Incheon 22711, Korea; imkhy77@gmail.com; 2Sleep Disorders Center, Department of Neurology, Soonchunhyang University, College of Medicine, Cheonan Hospital, Cheonan 31151, Korea; jkpark.oscillator@schmc.ac.kr; 3Department of Neurology, Kangbuk Samsung Hospital, Seoul 03181, Korea; ultrajs4@gmail.com; 4Clinical Trial Center, Department of Biostatics, Soonchunhyang University, Cheonan 31151, Korea; hello3933@schmc.ac.kr; 5Department of Neurology, Ewha Womans University Seoul Hospital, Ewha Womans University, Seoul 07804, Korea; fever26@gmail.com; 6Department of Neurology, Samsung Medical Center, Samsung Advanced Institute for Health Sciences & Technology, School of Medicine, Sungkyunkwan University, Seoul 06351, Korea

**Keywords:** restless legs syndrome, periodic limb movement during sleep, adolescents, daytime sleepiness, sleep habit, cross-sectional study

## Abstract

Although periodic limb movement during sleep (PLMS) and restless legs syndrome (RLS) are common in children and adolescents, they are relatively overlooked as a target of treatment. PLMS has been evaluated as having a lower clinical significance than RLS. This study examined the relationship of socio-behavioral factors of PLMS in Korean adolescents and checked whether PLMS is associated with excessive daytime sleepiness (EDS), depending on whether restless legs symptoms accompany it. In a cross-sectional study, 25,789 adolescents between 12 and 18 years of age (15.76 ± 1.73 years; female 51.49%) were evaluated using an online survey. Various self-report questionnaires were used to assess PLMS and RLS symptoms, EDS, sleep habits, and various socio-behavioral factors. The prevalence of self-perceived PLMS and restless legs symptoms were 903 (3.50%) and 1311 (5.08%), respectively. Of the 1311 participants, 399 had self-perceived PLMS. The odds ratios (ORs) for self-perceived PLMS in participants with restless legs symptoms were: males (OR = 1.528; 95% CI: 1.145–2.040), usually/always experienced apnea apnea (OR, 3.006; 95% CI, 1.954–4.624), increased proneness to Internet addiction (OR = 1.013; 95% CI: 1.001–1.025), sometimes/often consuming coffee (OR = 1.312; 95% CI: 1.015–1.695), EDS (OR = 0.826; 95% CI: 0.488–1.398), and perceived insufficient sleep (OR = 1.143; 95% CI: 0.835–1.565). The male gender, witness apnea, consuming coffee, and being prone to Internet addiction were identified as factors significantly associated with self-perceived PLMS in participants with restless legs symptoms. However, EDS and insufficient sleep were associated with self-perceived PLMS in the absence of restless legs symptoms.

## 1. Introduction

Periodic limb movement during sleep (PLMS), a clinically observed finding and frequent motor phenomena observed in polysomnography (PSG) recording, has a reported prevalence of about 4–11% in adults and 1–2% in children [1]. It is described as a stereotypical involuntary movement during sleep, often occurring in patients with restless legs syndrome (RLS) [2]. PLMS is a clinical symptom commonly observed in persons with sleep apnea syndrome, rapid eye movement (REM) sleep behavior disorder, and narcolepsy [3]. In particular, presenting more than 15 periodic limb movements per hour is related to daytime sleepiness due to low sleep quality and is frequently reported in elderly patients with RLS [2]. However, PLMS has been commonly observed among elderly and healthy adults without symptoms of RLS [4]. According to a general population study in German adults, age, the male gender, RLS, physical inactivity, current smoking, diabetes, antidepressant use, and lower serum magnesium were reported as risk factors of PLMS [5]. Compared to RLS risk factors, epidemiologic studies on PLMS are lacking. In addition, PLMS has been evaluated as having a lower clinical significance than RLS. The opinions about the clinical significance of PLMS vary, with some views deeming sleep disorders with PLMS as more clinically significant than PLMS itself. Furthermore, although PLMS is closely related to RLS, it is necessary to approach the two conditions separately when evaluating the clinical symptoms and influencing factors of PLMS [6].

### Aims of the Present Study

Although PLMS has been observed in a PSG study, few studies have focused on the clinical significance of PLMS as a symptom itself. This study investigated the relationship between socio-behavioral factors pertaining to self-perceived PLMS in Korean adolescents, and examined whether self-perceived PLMS is associated with excessive daytime sleepiness (EDS), depending on whether restless legs symptoms accompanies it.

## 2. Materials and Method

### 2.1. Study Design and Participants

This nationwide cross-sectional study was conducted from 1 July to 31 July 2011, by the Sleep Center at the Samsung Medical Center and Korea Centers for Disease Control and Prevention. It evaluated the sleep health status of Korean adolescents aged 12–18 years old and its effects on their daily lives by using an online survey. The nationwide random selection of 75 middle schools and 75 high schools (7th–12th school grades) and collection of students’ information was supported by the Korean Ministry of Education, Science, and Technology. Two classes were randomly selected from each grade level of each school to represent the overall population of that grade. This study was approved by the Institutional Review Board of the Samsung Medical Center, Seoul, Korea.

### 2.2. Demographics

The demographic variables included age, sex, school grade, and body mass index (BMI, calculated from body weight and height). The BMI z-score was adjusted for age and sex, and was estimated according to the 2017 Korean National Growth Charts for children and adolescents.

### 2.3. Measurement of Symptoms of RLS and Self-Perceived PLMS

The self-perceived PLMS and restless legs were evaluated using the Global Sleep Assessment Questionnaire (GSAQ) [7]. The questionnaire consists of 11 questions on insomnia disorders, insomnia associated with a mental disorder, obstructive sleep apnea, RLS, and PLMS. Questions included: “Over the past four weeks, did you have any episodes of restlessness, or a crawling feeling in your legs, at night that went away if you moved your legs?” for symptoms of RLS, and “Did you have any episodes of repeated rhythmic jerkiness in your legs, or twitchy legs, during your sleep?” for self-perceived PLMS. Based on the answers to these questions of frequency (i.e., questions answerable by “never”, “sometimes”, “usually”, or “always”), the respective answers “usually” or “always” indicated the presence of either restless legs symptoms or self-perceived PLMS.

### 2.4. Other Sleep-Related Measures

Daytime sleepiness was assessed using the Korean version of the Epworth Sleepiness Scale (ESS), a questionnaire consisting of eight items that quantifies the likelihood of dozing in eight different situations [8]. The last situation on the ESS, “in a car while stopped in traffic,” was originally for drivers. However, because persons younger than 20 years old are not allowed to drive in Korea, the question for this situation was modified for passengers instead of drivers. The total ESS score ranges from 0 to 24, where scores of 11–24 were defined as excessive daytime sleepiness in this study. Sleep characteristics also included snoring, experienced apnea apnea, and perceived sleep sufficiency.

### 2.5. Socio-Behavioral Factors

The risk of Internet addiction was assessed using the Internet Addiction Proneness Scale for Youth [9]. Other social behaviors were also surveyed, including the frequency of coffee and alcohol consumption, smoking, nocturnal eating, co-sleeping with a housemate, co-sleeping with a doll or pet, and sleeping with a television/radio on during sleep.

### 2.6. Statistical Analyses

Comparisons between participants with PLMS symptoms and without PLMS symptoms were performed using the t-test or chi-square test according to the nature of the variables. Logistic regression analyses were used to examine the odds ratios (OR) and 95% confidence intervals (CI) for the presence of PLMS symptoms after adjusting for relevant covariates. All statistical analyses were performed using SPSS Statistics ver. 26.0 (IBM Corp., Armonk, NY, USA).

## 3. Results

### 3.1. Descriptive Demographics

In total, 25,789 students (13,340 middle school and 12,449 high school students) with a mean age of 15.76 ± 1.73 years (mean ± standard deviation) participated in this study. Among them, 51.49% (*n* = 13,279) were female and 3.50% (*n* = 903) presented self-perceived PLMS. In addition, the participants’ average BMI was 20.70 ± 3.27.

### 3.2. Characteristics of Participants with Self-Perceived PLMS

Table 1 presents the participants’ characteristics according to the presence of self-perceived PLMS. Such symptoms were significantly prominent in males (*n* = 500, 55.37%; *p* < 0.001) and participants aged approximately 16.11 ± 1.60 years (*p* < 0.001). Those who answered usually/always to the self-perceived PLMS question were older than those who answered never/sometimes. In addition, participants with the presence of self-perceived PLMS usually/always snored, usually/always experienced sleep apnea, had high BMI, sometimes/often consumed coffee and alcohol, sometimes/often smoked, often participated in nocturnal eating, co-slept with a doll or pet, played television/radio during sleep, usually/always suffered RLS symptoms, and were prone to Internet addiction (*p* < 0.001). Moreover, participants with self-perceived PLMS were more likely to perceive insufficient sleep and experience increased daytime sleepiness (*p* < 0.001).

### 3.3. Characteristics of Participants with Self-Perceived PLMS in Participants with Restless Leg Symptoms

Table 2 presents the participants’ characteristics according to self-perceived PLMS in participants with restless legs symptoms. The average age of participants with restless legs and self-perceived PLMS (16.25 ± 1.60) was slightly older than participants with restless legs symptoms and absence of self-perceived PLMS (15.74 ± 1.67) (*p* < 0.001). However, there was no significant difference concerning the participants’ sex. In addition, results showed that participants with usually/always PLMS symptoms and restless legs symptoms usually/always snored (*p* = 0.003), usually/always experienced sleep apnea (*p* < 0.001), ESS (*p* = 0.001), sometimes/often consumed coffee and alcohol (*p* < 0.001), often smoked (*p* = 0.002), often participated in nocturnal eating (*p* < 0.001), and were prone to Internet addiction (*p* = 0.002). Moreover, although these participants usually/always suffering from PLMS symptoms and restless legs symptoms showed perceived insufficient sleep (*p* = 0.009), the EDS did not detect significant differences from participants with restless legs symptoms and absence of self-perceived PLMS (*p* = 0.383).

### 3.4. Logistic Regression Model for the Presence of Self-Perceived PLMS in Participants with Restless Legs Symptoms

Table 3 shows the results of the adjusted odds ratios for the presence of self-perceived PLMS in participants with restless legs symptoms obtained from multivariate logistic regression analyses. The analyses revealed that those who usually/always experienced self-perceived PLMS were significantly associated with the following factors: males (OR = 1.528; 95% CI: 1.145–2.040), usually/always experienced apnea (OR, 3.006; 95% CI, 1.954–4.624), were prone to Internet addiction (OR = 1.013; 95% CI: 1.001–1.025), and sometimes/often consumed coffee (OR = 1.312; 95% CI: 1.015–1.695). In contrast, EDS (OR = 0.826; 95% CI: 0.488–1.398) and insufficient sleep (OR = 1.143; 95% CI: 0.835–1.565) were not associated with those who usually/always experienced self-perceived PLMS in this group.

### 3.5. Characteristics of Participants with Self-Perceived PLMS in Participants in the Absence of Restless Legs Symptoms

Table A1 shows the participants’ (*n* = 504, 2.0%) characteristics according to self-perceived PLMS in participants in the absence of restless legs symptoms. Moreover, the following factors were seen to be significantly associated with self-perceived PLMS: male, age, frequencies of experienced apnea apnea, snoring, perceived insufficient sleep, excessive daytime sleep (EDS), Internet addiction proneness, alcohol, smoking, nocturnal eating, sleeping with a television/radio on, and co-sleeping with a doll or pet. Furthermore, social behaviors such as Internet addiction proneness, coffee consumption, smoking, co-sleeping with a doll or pet, sleeping with a television/radio on, and nocturnal eating were more frequently reported in usually/always self-perceived PLMS without restless leg symptoms (*p* < 0.001).

### 3.6. Logistic Regression Model for Self-Perceived PLMS in Participants in the Absence of Restless Legs Symptoms

Table A2 shows the adjusted ORs for perceived PLMS in participants in the absence of restless legs symptoms obtained from multivariate logistic regression analyses. The analysis results revealed that perceived PLMS were significantly associated with 3rd grade middle school (OR = 2.010; 95% CI: 1.211–3.335) and 1st grade high school (OR = 2.398; 95% CI: 1.245–4.619). Furthermore, participants who were male (OR = 1.343; 95% CI: 1.095–1.646), usually/always snore (OR = 2.024; 95% CI: 1.479–2.770), sometimes (OR = 2.001; 95% CI: 1.505–2.660) and usually/always experienced apnea (OR = 4.268; 95% CI: 2.796–6.513), EDS (OR = 1.704; 95% CI: 1.174–2.475), and Internet addiction proneness (OR = 1.184; 95% CI: 0.978–1.433) showed significant associations with perceived PLMS.

## 4. Discussion

Self-perceived PLMS showed in approximately 3.5% of all participants in the present study. The prevalence of PLMS varies depending on population’s age, a study’s diagnostic tools, and its working definition for the terms used [10,11,12,13]. A prevalence survey was conducted in a single pediatric hospital by Kirk and Bohn [1], and reported that PLMS recorded by polysomnographic findings was infrequent in children, having a 1.2% prevalence rate. The results of the Korean pediatric RLS study by Kim et al. [14] showed participants were less likely to experience PLMS than adults.

Being male and experience of apnea were significantly associated with self-perceived PLMS, regardless of whether restless legs symptoms accompanied PLMS in the present study. Holzknecht et al. [12] found that PLMS was more prevalent in male patients with RLS than female patients with RLS. However, the causative etiology of higher PLMS prevalence in the male gender is unknown. Several studies have suggested that snoring and sleep apnea are more common in men, thus relating to their proneness to PLMS [15,16], yet such theories are considered controversial. Because sleep apnea and PLMS are closely related [17], previous studies indicate PLMS’s link to low respiratory arousal threshold in obstructive sleep apnea (OSA) [18]. In contrast, Iriarte et al. [19] found that although PLMS alters sleep structure, the relationship between PLMS and the sleep structure of patients with OSA was less significant. The subjects typically targeted in the PLMS and OSA studies have mainly been adults. Since OSA in adolescents has a different and/or more complex pathophysiology that that of adults, further consideration will be needed in order to apply the link to adolescents. As such, the correlation between OSA and PLMS in adolescents requires more research.

We also found a significant association between proneness to Internet addiction and self-perceived PLMS. In a study on children and adolescents, Internet addiction was reported to induce sleep disorders such as insomnia and circadian rhythm disorders [20,21]. Similarly, an association between insomnia and PLMS has been reported [22]. Internet addiction is expected to disrupt circadian rhythms, and insomnia is considered a symptom of a circadian disorder. However, the mutual relationship between insomnia, Internet addiction, and PLMS is not fully recognized. Internet addiction and a decrease in peripheral/striatal blood dopamine levels have been found to be linked in several studies [23,24]. In addition, the relationship between the dysfunction of the brain dopaminergic system and Internet addiction has been proven through nuclear imaging findings [25]. Furthermore, the mechanism of PLMS is also understood as a dysfunction of the dopaminergic system. These findings from functional images and cases provide evidence that dopamine administration improves these symptoms [26,27,28]. Subsequently, excessive sedentary behavior is common in adolescent Internet addiction [29], and is also associated with sleep disturbance [30]. Thus, proper leg exercises and reduced sedentary behavior have been suggested as treatments for PLMS [31].

Similarly, caffeine consumption was presented as another related factor for self-perceived PLM with restless legs symptoms. These caffeine and Internet addictions are well-known causes of sleep disturbance in adolescents [32]. However, the relationship between Internet addiction or caffeine consumption and PLMS has yet to be reported. Some studies have reported that Internet addiction and excessive caffeine consumption are related to adolescent depressive anxiety [32], and that adolescent PLMS is more frequently observed in adolescents with anxiety and depression [33]. Caffeine is also a well-known cause of RLS [34]. Based on those results, as a non-pharmacological treatment for RLS and PLMS, it is recommended to reduce caffeine intake, smoking, and drinking [35]. Although it is difficult to justify a causal relationship based on the results of this cross-sectional study alone, it can be asserted that PLMS symptoms, caffeine intake, and Internet addiction proneness mutually influence one another.

In participants reporting instances of restless leg symptoms, there was no statistically significant correlation found between EDS and self-perceived PLM. However, in participants with restless leg symptoms, self-perceived PLM was statistically correlated with EDS/perceived sleep insufficiency. Leary et al. [36] reported that daytime sleepiness is not associated with PLMS in patients with/without RLS; however, the severity of daytime sleepiness was associated with RLS without PLMS. In the PSG-based study of Shin et al. [13], the PLMS index showed no correlation with ESS in adult patients with RLS. Although this study is based on adolescents with self-perceived PLM, it showed similar results to previous studies based on PSG findings. This result implies that PLMS had little effect on EDS in patients with RLS symptoms because EDS is already closely associated with RLS. However, as seen in the participants’ absence of restless legs symptoms, self-perceived PLMS may be associated with insufficient sleep and EDS. PLMS should be recognized as also being of clinical significance in the absence of RLS.

This study has several limitations. First, it was conducted based on a questionnaire about perceived PLMS and restless legs symptoms. Therefore, it was challenging to compare the results with the previously reported PLMS prevalence based on PSG findings. Second, as a retrospective analysis of a questionnaire-based cohort study, questionnaires on medication and medical and psychiatric diseases known to affect PLMS were not included in this study. Third, this study was conducted using a Global Sleep Assessment Questionnaire (GASQ); items and questions pertaining to the self-perceived PLMS and RLS were described above. Although the questions are relatively simple, the severity and aspects of the symptoms may vary depending on the respondent, so there is a possibility that cases that mimic the RLS and PLM may be included. Therefore, to avoid confusion with PLMS, we introduced ‘self-perceived PLM’ as a new term, although further research using PSG will be needed. Fourth, the age of adolescents was clearly defined by the researchers. Considering that the WHO defines adolescents as 10–18 years of age and the American Academy of Pediatrics defines adolescents as 12–20 years of age, and because the current study was conducted at a school level, we compromised, with students between 12–18 years of age ultimately participating. One additional limitation in sampling is that students who did not attend school were excluded from the study. Lastly, in this study, the sleep duration of the participants was not analyzed. It has been discussed that the short sleep duration of Korean adolescents has caused many problems through several previous studies [37,38]. However, this study excluded the consideration of sleep duration because it focused on the socio-behavioral factors related to PLM symptoms. Despite these limitations, however, this study is meaningful not only as a large-scale study, but also as a study examining PLMS as a clinical symptom.

## 5. Conclusions

In conclusion, being male, experience of apnea, consumption of caffeine, and proneness to Internet addiction were identified as factors significantly associated with self-perceived PLMS in participants with restless legs symptoms. Self-perceived PLMS was not associated with EDS and insufficient sleep in adolescents with restless legs symptoms, but was found an association in adolescents without restless legs symptoms. Although this was a questionnaire-based study of perceived symptoms of PLMS, the findings are comparable with existing studies based on PSG results.

## Figures and Tables

**Table 1 ijerph-19-04751-t001:** Participants’ characteristics according to the presence of self-perceived PLMS.

Characteristic	Presence	Absence	Total	*p*-Value
*n* (%)	903 (3.50)	24,886 (96.50)	25,789 (100.00)	
Grade level, *n* (%)				<0.001
M1	92 (10.19)	4354 (17.50)	4446 (17.24)	
M2	121 (13.40)	4380 (17.60)	4501 (17.45)	
M3	160 (17.72)	4233 (17.01)	4393 (17.03)	
H1	201 (22.26)	4179 (16.79)	4380 (16.98)	
H2	178 (19.71)	4002 (16.08)	4180 (16.21)	
H3	151 (16.72)	3738 (15.02)	3889 (15.08)	
Sex, *n* (%)				<0.001
Female	403 (44.63)	12,876 (51.74)	13,279 (51.49)	
Male	500 (55.37)	12,010 (48.26)	12,510 (48.51)	
Age, mean ± SD	16.11 ± 1.60	15.75 ± 1.74	15.76 ± 1.73	<0.001
BMI, mean ± SD	21.12 ± 3.51	20.69 ± 3.26	20.70 ± 3.27	<0.001
Category, *n* (%)				0.227
Underweight	66 (7.31)	1863 (7.49)	1929 (7.48)	
Normal	686 (75.97)	19,447 (78.14)	20,133 (78.07)	
Overweight	73 (8.08)	1821 (7.32)	1894 (7.34)	
Obese	78 (8.64)	1755 (7.05)	1833 (7.11)	
Snored, *n* (%)				<0.001
Never	576 (63.79)	20,067 (80.64)	20,643 (80.05)	
Sometimes	177 (19.60)	3673 (14.76)	3850 (14.93)	
Usually/Always	150 (16.61)	1146 (4.60)	1296 (5.03)	
Experienced apnea, *n* (%)				<0.001
Never	633 (70.10)	22,990 (92.38)	23,623 (91.60)	
Sometimes	167 (18.49)	1595 (6.41)	1762 (6.83)	
Usually/Always	103 (11.41)	301 (1.21)	404 (1.57)	
Perceived sleep sufficiency				<0.001
Sufficient	208 (23.03)	9388 (37.72)	9596 (37.21)	
Much	37 (4.10)	1525 (6.13)	1562 (6.06)	
Insufficient	658 (72.87)	13,973 (56.15)	14,631 (56.73)	
ESS, mean ± SD	17.07 ± 4.93	14.25 ± 3.87	14.35 ± 3.95	<0.001
EDS				<0.001
ESS < 11, *n* (%)	58 (6.42)	4001 (16.08)	4059 (15.74)	
ESS ≥ 11, *n* (%)	845 (93.58)	20,885 (83.92)	21,730 (84.26)	
Internet addiction proneness, mean ± SD	32.45 ± 11.68	28.21 ± 8.06	28.36 ± 8.25	<0.001
Coffee				<0.001
Never	470 (52.05)	16,280 (65.42)	16,750 (64.95)	
Sometimes	328 (36.32)	7324 (29.43)	7652 (29.67)	
Often	105 (11.63)	1282 (5.15)	1387 (5.38)	
Alcohol				<0.001
Never	737 (81.62)	23,009 (92.46)	23,746 (92.08)	
Sometimes	134 (14.84)	1794 (7.21)	1928 (7.48)	
Often	32 (3.54)	83 (0.33)	115 (0.45)	
Smoking				<0.001
Never	766 (84.83)	23,270 (93.51)	24,036 (93.20)	
Sometimes	54 (5.98)	610 (2.45)	664 (2.57)	
Often	83 (9.19)	1006 (4.04)	1089 (4.22)	
Nocturnal eating				<0.001
Never	147 (16.28)	6180 (24.83)	6327 (24.53)	
Sometimes	485 (53.71)	13,724 (55.15)	14,209 (55.10)	
Often	271 (30.01)	4982 (20.02)	5253 (20.37)	
Co-sleeping with housemate				0.283
With	314 (34.77)	9089 (36.52)	9403 (36.46)	
Without	589 (65.23)	15,797 (63.48)	16,386 (63.54)	
Co-sleeping with doll or pet				< 0.001
With	211 (23.37)	4213 (16.93)	4424 (17.15)	
Without	692 (76.63)	20,673 (83.07)	21,365 (82.85)	
Television/Radio during sleep				<0.001
On	135 (14.95)	2326 (9.35)	2461 (9.54)	
Off	768 (85.05)	22,560 (90.65)	23,328 (90.46)	
Restless legs symptoms				<0.001
Never/Sometimes	504 (55.81)	23,974 (96.34)	24,478 (94.92)	
Usually/Always	399 (44.19)	912 (3.66)	1311 (5.08)	

Abbreviations: PLMS, periodic limb movement during sleep; M1–M3, middle school 1st–3rd grade; H1–H3, high school 1st–3rd grade; SD, standard deviation; ESS, Epworth Sleepiness Scale; EDS, excessive daytime sleep; BMI, body mass index.

**Table 2 ijerph-19-04751-t002:** Participants’ characteristics according to the presence of self-perceived PLMS in participants with restless legs symptoms.

Characteristic	Presence	Absence	Total	*p*-Value
*N*	399	912	1311	
Grade level, *n* (%)				<0.001
M1	38 (9.52)	158 (17.32)	196 (14.95)	
M2	51 (12.78)	155 (17.00)	206 (15.71)	
M3	63 (15.79)	166 (18.20)	229 (17.47)	
H1	83 (20.80)	167 (18.31)	250 (19.07)	
H2	92 (23.06)	141 (15.46)	233 (17.77)	
H3	72 (18.05)	125 (13.71)	197 (15.03)	
Sex, *n* (%)				0.081
Female	166 (41.60)	427 (46.82)	593 (45.23)	
Male	233 (58.40)	485 (53.18)	718 (54.77)	
Age, mean ± SD	16.25 ± 1.60	15.74 ± 1.67	15.89 ± 1.67	0.001
BMI, mean ± SD	21.30 ± 3.67	20.88 ± 3.49	21.00 ± 3.55	0.048
Category, *n* (%)				0.755
Underweight	26 (6.52)	67 (7.35)	93 (7.09)	
Normal	300 (75.19)	693 (75.99)	993 (75.74)	
Overweight	33 (8.27)	76 (8.33)	109 (8.31)	
Obese	40 (10.03)	76 (8.33)	116 (8.85)	
Snored, *n* (%)				0.003
Never	225 (56.39)	597 (65.46)	822 (62.70)	
Sometimes	89 (22.31)	182 (19.96)	271 (20.67)	
Usually/Always	85 (21.30)	133 (14.58)	218 (16.63)	
Experienced apnea, *n* (%)				<0.001
Never	236 (59.15)	686 (75.22)	922 (70.33)	
Sometimes	93 (23.31)	168 (18.42)	261 (19.91)	
Usually/Always	70 (17.54)	58 (6.36)	128 (9.76)	
Perceived sleep sufficiency				0.009
Sufficient	76 (19.05)	224 (24.56)	300 (22.88)	
Much	11 (2.76)	46 (5.04)	57 (4.35)	
Insufficient	312 (78.20)	642 (70.39)	954 (72.77)	
ESS, mean ± SD	18.11 ± 5.37	16.52 ± 4.59	17.00 ± 4.90	0.001
EDS				0.383
ESS < 11, *n* (%)	24 (6.02)	67 (7.35)	91 (6.94)	
ESS ≥ 11, *n* (%)	375 (93.98)	845 (92.65)	1220 (93.06)	
Internet addiction proneness, mean ± SD	33.86 ± 13.12	31.56 ± 9.95	32.26 ± 11.06	0.002
Coffee				<0.001
Never	190 (47.62)	550 (60.31)	740 (56.45)	
Sometimes	155 (38.85)	286 (31.36)	441 (33.64)	
Often	54 (13.53)	76 (8.33)	130 (9.92)	
Alcohol				<0.001
Never	308 (77.19)	796 (87.28)	1104 (84.21)	
Sometimes	67 (16.79)	109 (11.95)	176 (13.42)	
Often	24 (6.02)	7 (0.77)	31 (2.36)	
Smoking				0.002
Never	327 (81.95)	812 (89.04)	1139 (86.88)	
Sometimes	22 (5.51)	31 (3.40)	53 (4.04)	
Often	50 (12.53)	69 (7.57)	119 (9.08)	
Nocturnal eating				<0.001
Never	60 (15.04)	199 (21.82)	259 (19.76)	
Sometimes	198 (49.62)	488 (53.51)	686 (52.33)	
Often	141 (35.34)	225 (24.67)	366 (27.92)	
Co-sleeping with housemate				0.602
With	257 (64.41)	601 (65.90)	858 (65.45)	
Without	142 (35.59)	311 (34.10)	453 (34.55)	
Co-sleeping with doll or pet				0.157
With	105 (26.32)	207 (22.70)	312 (23.80)	
Without	294 (73.68)	705 (77.30)	999 (76.20)	
Television/radio during sleep				0.140
On	71 (17.79)	133 (14.58)	204 (15.56)	
Off	328 (82.21)	779 (85.42)	1107 (84.44)	

Abbreviations: PLMS, periodic limb movement during sleep; M1–M3, middle school 1st–3rd grade; H1–H3, high school 1st–3rd grade; SD, standard deviation; ESS, Epworth Sleepiness Scale; EDS, excessive daytime sleep; BMI, body mass index.

**Table 3 ijerph-19-04751-t003:** Odds ratios for the presence of self-perceived PLMS in participants with restless legs symptoms.

Characteristic	OR (95% CI)	*p*-Value
Grade		
M1	Ref.	
M2	1.205 (0.695–2.087)	0.506
M3	1.190 (0.596–2.374)	0.622
H1	1.368 (0.573–3.266)	0.481
H2	1.612 (0.547–4.752)	0.387
H3	1.287 (0.341–4.866)	0.710
Sex		
Female	Ref.	
Male	1.528 (1.145–2.040)	0.004
Age	1.118 (0.865–1.445)	0.396
BMI		
Underweight	Ref.	
Normal	1.173 (0.714–1.929)	0.529
Overweight	1.229 (0.644–2.345)	0.531
Obese	1.204 (0.637–2.276)	0.569
Snored		
Never	Ref.	
Sometimes	1.095 (0.800–1.499)	0.571
Usually/Always	1.107 (0.763–1.606)	0.593
Experienced apnea		
Never	Ref.	
Sometimes	1.421 (1.036–1.949)	0.029
Usually/Always	3.006 (1.954–4.624)	<0.001
Perceived sleep sufficiency		
Sufficient	Ref.	
Insufficient	1.143 (0.835–1.565)	0.404
EDS		
ESS < 11	Ref.	
ESS ≥ 11	0.826 (0.488–1.398)	0.476
Internet addiction proneness	1.013 (1.001–1.025)	0.031
Coffee		
Never	Ref.	
Sometimes/Often	1.312 (1.015–1.695)	0.038
Alcohol		
Never	Ref.	
Sometimes/Often	1.302 (0.882–1.923)	0.184
Smoking		
Never	Ref.	
Sometimes/Often	1.067 (0.690–1.651)	0.771
Nocturnal eating		
Never	Ref.	
Sometimes/Often	1.328 (0.942–1.872)	0.106
Co-sleeping with housemate		
With	1.103 (0.847–1.436)	0.465
Without	Ref.	
Co-sleeping with doll or pet		
With	1.074 (0.793–1.456)	0.643
Without	Ref.	
Television/radio during sleep		
On	1.092 (0.781–1.526)	0.606
Off	Ref.	

Abbreviations: PLMS, periodic limb movement during sleep; Ref., reference; M1–M3, middle school 1st–3rd grade; H1–H3, high school 1st–3rd grade; SD, standard deviation; ESS, Epworth Sleepiness Scale; EDS, Excessive Daytime Sleep; BMI, body mass index.

## Data Availability

The data that support the findings of this study are available on request from the corresponding author [KIY].

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
