# Peer review of "Association between Self-Perceived Periodic Limb Movement during Sleep and Excessive Daytime Sleepiness Depend on Restless Leg Symptoms in Korean Adolescents"

_ijerph, 2022, doi:10.3390/ijerph19084751_

Round 1
Reviewer 1 Report
Kim et al. described a large scale study to investigate the relationship between PLMS and daytime sleepiness in the presence and absence of RLS. The most striking concern about the study is the use of self-reported questionnaire to assess the most crucial variables, presence/absence of PLMS and daytime sleepiness. To the credit of the authors, they acknowledged this limitation in the discussion section. However, it may be worth to cross validate this association in a publicly available quantitative dataset (from PSG, for example), if possible.
Secondly, the authors should perform a thorough check on English language and sentence construction. There are some places that need re-working. One such example is line 244 - "In our study, insufficiency sleep and EDS were statistically insignificant related with self-perceived PLMS in restless legs symptom" should read "In our study, relationship between insufficient sleep and EDS with self perceived PLMS in RLS was statisticallt insignificant".
Author Response
Dear reviewer
Thank you for your warm and thoughtful review.
We have added our answers point by point with attached file.
Thank you for your comment, which is very helpful for our article to develop further.

Reviewer 2 Report
The authors assessed PLMS by using a questionnaire - GSAQ (Global Sleep Assessment Questionnaire). GSAQ is a tool for screening for potential sleep disorders, mostly in primary care settings.
These are the diagnostic criteria for assessment of PLMS by American Academy of Sleep Medicine
Diagnostic criteria for periodic limb movement disorderPolysomnography shows repetitive, highly stereotyped limb movements that are: |
|
|
|
|
The PLMS index is ≥5/hour in pediatric cases and ≥15/hour in adult cases |
PLMS cause clinical sleep disturbance (difficulty with sleep initiation, sleep maintenance, and/or unrefreshing sleep) or impaired daytime function* |
The PLMS are not better explained by another current sleep disorder¶, medical or neurologic disorder, mental disorder, medication use, or substance use disorder (eg, exclude from PLMS counts the movements at the termination of cyclically occurring apneas) |
PLMS can be diagnosed only by PSG. GSAQ can only reveal potential sleep disorder that has to be further assessed. The study should focus on other sleep disorders that can be self-assessed as RLS or insomnia.
Author Response

(The authors gave the same response as above.)

Reviewer 3 Report
The aim oft he study was to examine the self perceived PLMS/RLS and socio-behavioral factors and their relationship to excessive daytime sleepiness. This very large survey of high school students used the GASQ which has been frequently used to investigate sleep disorders in large populations, but which has a not very high sensitivity and specificity. The one question for RLS and PLMS only make it difficult to give an estimate of the real prevalence of the respective disorders.
Due to the validity of the test the introduction should give some insights into the problems that this test has to confirm sleep disorders.
Line 65 should be: …and examine if….instead of ….examine that….
Line 72: What was the rationale to choose adolescents?
Line 90: possible mimics of RLS and PLMS should at least be included in the discussion section (i.e. habitual repetitive leg movements due to inner tension)
Line 105: authors should explain why they focus so much on internet addiction. Is this a major problem of young Koreans?
Line 150: persons with self perceived PLMS and RLS had significantly more self perceived insufficient sleep, but no increased daytime sleepiness. This is astonishing and could be due to lack of operationalization of the self-perceived problems. Table 2 shows that mean ESS is sign. higher in the „present“ group. Please comment.
Table 2: it is amazing how many participants had an ESS score >11 in both groups. Is this related to sleep deprivation or short sleep? It is also astonishing that so many sleep with a housemate or a pet/doll? Ist hat still ages specific or specific for a Korean population of this age? Insufficient sleep is extremely high in both groups. Please comment.
The discussion should always take into account that this survey was performed in adolescents and should retain to generalize conclusions. This is specifically tricky for self reported apneas with sleep apnea being rare in adolescents.
Author Response

(The authors gave the same response as above.)

Round 2
Reviewer 3 Report
There are several points the authors did not respond to adequately, namely:
- short sleep and daytime sleepiness: there is much literature about this topic, which was not referred to by authors. Outside of Korea it is not known that Korean adolescents sleep little. please show a reference
- The high prevalence of internet addiction is not known outside of Korea. Please give a literature citation
- it is not known that Korean adolescents sleep with pets. Any citation for that?
Author Response
Dear Reviewer:
Thank you for your warm and careful review.
Details are provided in the attached file. I agree with your comment, and I have added and revised the content.
